# Food Waste from Campus Dining Hall as a Potential Feedstock for 2,3-Butanediol Production via Non-Sterilized Fermentation

**DOI:** 10.3390/foods13030452

**Published:** 2024-01-31

**Authors:** Alicia Caldwell, Xueqian Su, Qing Jin, Phyllicia Hemphill, Doaa Jaha, Sonecia Nard, Venkataswarup Tiriveedhi, Haibo Huang, Joshua OHair

**Affiliations:** 1Department of Biological Sciences, College of Life & Physical Sciences, Tennessee State University, Nashville, TN 37209, USA; aliciacaldwell97@gmail.com (A.C.); phylliciahemphill@gmail.com (P.H.); djaha@my.tnstate.edu (D.J.); sonecianard@gmail.com (S.N.); vtirivee@tnstate.edu (V.T.); 2Department of Food Science and Technology, College of Agriculture & Life Sciences, Virginia Tech, Blacksburg, VA 24061, USA; xueqians@vt.edu (X.S.); huang151@vt.edu (H.H.); 3School of Food and Agriculture, College of Earth, Life, and Health Sciences, University of Maine, Orono, ME 04469, USA; qing.jin@maine.edu

**Keywords:** food waste, microbial fermentation, 2,3-butanediol, 2,3-BD, 2,3-BDO, thermophiles, biofuel, bioproducts, renewable feedstock, non-sterile, non-model organism

## Abstract

Food waste is a major issue that is increasingly affecting our environment. More than one-third of food is wasted, resulting in over $400 billion in losses to the U.S. economy. While composting and other small recycling practices are encouraged from person-to-person, it is not enough to balance the net loss of 80 million tons per year. Currently, one of the most promising routes for reducing food waste is through microbial fermentation, which can convert the waste into valuable bioproducts. Among the compounds produced from fermentation, 2,3-butanediol (2,3-BDO) has gained interest recently due to its molecular structure as a building block for many other derivatives used in perfumes, synthetic rubber, fumigants, antifreeze agents, fuel additives, and pharmaceuticals. Waste feedstocks, such as food waste, are a potential source of renewable energy due to their lack of cost and availability. Food waste also possesses microbial requirements for growth such as carbohydrates, proteins, fats, and more. However, food waste is highly inconsistent and the variability in composition may hinder its ability to be a stable source for bioproducts such as 2,3-BDO. This current study focuses specifically on post-consumer food waste and how 2,3-BDO can be produced through a non-model organism, *Bacillus licheniformis* YNP5-TSU during non-sterile fermentation. From the dining hall at Tennessee State University, 13 food waste samples were collected over a 6-month period and the compositional analysis was performed. On average, these samples consisted of fat (19.7%), protein (18.7%), ash (4.8%), fiber (3.4%), starch (27.1%), and soluble sugars (20.9%) on a dry basis with an average moisture content of 34.7%. Food waste samples were also assessed for their potential production of 2,3-BDO during non-sterile thermophilic fermentation, resulting in a max titer of 12.12 g/L and a 33% g/g yield of 2,3-BDO/carbohydrates. These findings are promising and can lead to the better understanding of food waste as a defined feedstock for 2,3-BDO and other fermentation end-products.

## 1. Introduction

### Unsterilized Food Waste as a Feedstock for Bioproducts

Food waste is a major issue that has long-lasting impacts on our environment. One study calculated the impact of food waste on 15 different countries and found that the U.S. environmental impact alone produced 172 million tons of CO_2_, consumed 22 million tons of oil, and used 11 billion cubic meters of water [1]. Food waste is generally categorized as such when it is either discarded or disposed of without being consumed [2]. Statistically, more than one-third of food is wasted, weighing about 1.3 billion metric tons, and equating to about a $750 billion loss to the global economy [3]. Food waste can occur at many different stages, including production, handling/storage, processing/packaging, distribution/market, and consumption [4]. In a study conducted by the EPA in 2021, they concluded that food waste is the single most common material landfilled and incinerated in the U.S., accounting for 24% of landfilled municipal solid waste [5]. By using the life cycle assessment (LCA) approach and comparing composting, anaerobic digestion, and incineration food waste management techniques, one study concluded that landfills are the least beneficial and most harmful choice when it comes to environmental impact [6]. Therefore, it is important to find alternative ways to reduce or repurpose food waste.

Currently, one of the most promising routes for reducing food waste is converting it into valuable bioproducts [7]. One way this can be carried out is through the natural process of fermentation. Some common end-products from fermentation include, propionic acid, butyric acid acetic acid, lactic acid, ethanol, and other small organic acids and depending on the genera, one or more of these end-products will be produced [8]. Bacteria and fungi are both used in the bio-fermentation industry with key genera including *Lactobacillus*, *Bacillus*, *Clostridium*, *Enterobacter*, *Acetobacter*, *Saccharomyces*, *Penicillium*, *and Rhizopus* [8]. Of these, *Bacillus* spp. are one of the most widespread and well-studied, as they can grow in a wide range of environmental conditions. *Bacillus* spp. are aerobic endospore-forming bacteria that secrete a vast array of bioactive metabolites, enzymes, antibiotics, and end-products that have potential industrial use [9]. *Bacillus* spp. are safe and most species have little or no pathogenic potential as many are considered GRAS (generally regarded as safe) strains, allowing for use in various food applications [10]. Among the bio-products produced by *Bacillus* spp., 2,3-butanediol (2,3-BDO) is one of the most popular. This 4-carbon, 2-hydroxyl molecule can be used as a drop-in fuel or a building block for many derivatives, such as 1,3-butadiene and methyl ethyl ketone, which are used in perfumes, synthetic rubber, fumigants, antifreeze agents, fuel additives, and pharmaceuticals [11]. Until recently 2,3-BDO was primarily chemically synthesized, but bio-based (microbially derived) 2,3-BDO has economic and environmental advantages that could make it the preferred method in the future [12].

One major downside to bio-based 2,3-BDO production is the production cost. To reduce costs, waste (renewable) feedstocks are preferred, and while food waste is one potential option, the natural microorganisms in food waste can interfere with 2,3-BDO fermentation [13]. This current study focuses specifically on post-consumer food waste collection and how 2,3-BDO can be produced by using a non-sterile fermentation approach. This differs from previous studies in which agro-industrial waste, along with food wastes, cheese, whey, dried fruit and vegetables [14], and bread waste [3] were used to produce bioproducts before reaching the consumer. Several waste agro-industrial feedstocks that have previously been used to produce 2,3-BDO are corn steep liquor [15], sugarcane bagasse [16], corn stover hydrolysate [17], apple pomace [11], lignocellulosic biomass [18], and sugarcane molasses [19,20]. Although these waste samples were converted to 2,3-BDO, these experiments implemented mesophilic bacteria which required the use of autoclaving to sterilize feedstocks and stock cultures. The sterilization process increases the time and energy needed prior to inoculation and potentially the profitability and sustainability of bio-based 2,3-BDO production [21]. To circumvent sterilization, thermophilic fermentation has been shown to be effective and involves higher fermentation temperatures (above 45 °C). One such thermophilic bacteria, *Bacillus licheniformis* YNP5-TSU [22], was shown in a previous experiment to produce yields of 0.31–0.48 g 2,3-BDO/g sugar from various food wastes (pepper, pineapple and miscellaneous wastes) [21] while also eliminating the sterilization step. However, wastes were from pre-consumer stages during refrigerated preparation of meals. It is currently unknown if the same yields can be maintained from post-consumer food waste, which can be highly inconsistent with unknown amounts of contaminating microorganisms.

## 2. Materials and Methods

### 2.1. Food Collection and Feedstock Preparation

Food waste was collected 1–2 times per week for a total of 13 samples. In the cafeteria, participants were encouraged to hand sort non-consumable items such as plastics, paper, and metals into standard waste bins before placing leftover food into designated receptacles (Figure 1A,B). This process reduced the amount of sorting prior to the blending of the raw food waste slurry. All food waste samples were taken from the Dining Hall (Floyd Payne Student Center) at Tennessee State University, Nashville, TN, between the months of February and July 2022. Samples were collected from both breakfast (8:00 a.m.–11:00 a.m.) and lunch (11:00 a.m.–3:00 p.m.) periods using specialized food waste bins and weighed on average 10–20 lbs. Food waste media was prepared by adding 50 g of food waste into 100 mL distilled H_2_O and blended in a Vitamix^®^ 510 series blender to create a homogenous mixture. After several attempts, we found this was the best consistency for the food waste slurry (Figure 1C). Following homogenizing the mixture was brought to a pH of 8.0 (preferred pH: *B. licheniformis* YNP5-TSU [23]) by adding a 4% NaOH solution. 

### 2.2. Determination of Moisture, Ash, Protein, Fat, Fiber, and Starch

The chemical composition of the raw food waste was analyzed according to the method described by He et al. (2019) [24]. Briefly, the moisture content was determined by oven-drying at 105 °C until a constant weight was achieved. The samples were subjected to incineration in a muffle furnace at 550 °C for 6 h to determine the ash content. The protein content was quantified using the Kjeldahl nitrogen analysis method [25] with a conversion factor of 6.25. The fat content was determined using the Soxtec Method [26] with petroleum ether as the extraction solvent. The ANKOM Filter Bag System (ANKOM 2000 automated fiber analyzer, ANKOM Technology, Macedon, NY, USA) was employed to measure the fiber content in the fermented samples. The starch content was determined using the HCl hydrolysis method as described by Vidal, Rausch, Tumbleson and Singh (2009) [27].

### 2.3. Culture Propagation and Fermentation

The food waste mixture (100 mL) was transferred to 250 mL flasks for batch fermentation to produce 2,3-BDO. A 5% inoculum of *B. licheniformis* YNP_5_-TSU stock culture at 0.8 OD was added to food waste media to initiate fermentation. Stock cultures were prepared from previously described methods [23] by preparing initial growth media (P_1_) of 20 g/L glucose, 10 g/L yeast extract, and 5 g/L peptone. This 100 mL of P_1_ media was inoculated with frozen stock and incubated at 50 °C at 150 rpm. After 24 h, 20 mL of P_1_ was added to 80 mL of a P_2_ medium (60 g/L glucose, 10 g/L yeast extract, and 5 g/L peptone). The inoculated P_2_ medium was incubated for 3 h at 50 °C at 150 rpm after which 5 mL of P_2_ was transferred to 95 mL of a P_3_ medium (60 g/L glucose, 10 g/L yeast extract, and 10 g/L peptone). The inoculated P_3_ was then incubated for 3 h at 50 °C and 150 rpm until the stock culture reached between 0.5 and 0.8 OD. Each food waste fermentation was repeated in triplicate. 

### 2.4. Quantification of 2,3-BDO and Free Sugars

After each fermentation period (0, 24, 48 and 72 h), 1 mL of the fermented samples was collected and centrifuged at 10,000 rpm for 10 min (Eppendorf© 5453 Minispin Plus Centrifuge). Then, the supernatant was filtered through a 0.20 μm syringe filter (Waters Corporation, Milford, MA, USA). A 250 μL aliquot of the supernatant was diluted (1:4 ratio) with 0.005 M H_2_SO_4_ to ensure the acidity of solution for downstream analysis. Quantification of 2,3-BDO and free sugars, namely glucose, sucrose, fructose, and raffinose, in the collected fermentation samples was performed using high-performance liquid chromatography (HPLC), equipped with a refractive index detector (Agilent Technologies 1260, Santa Clara, CA, USA). Bio-Rad Aminex^®^ HPX-87H ion exclusion column (Bio-Rad Laboratories, Hercules, CA, USA) was employed to separate the compound of interest using 0.005 M H_2_SO_4_ as the mobile phase with a flow rate of 0.6 mL/min at 50 °C. The injection volume was 5 μL and the total running time was 30 min. All samples were analyzed in triplicate.

### 2.5. Statistical Analysis

The compositional analyses of food waste and 2,3-BDO fermentation samples of each food waste sample were conducted in duplicate. The mean, standard deviation and mass balance totals were calculated using statistical analysis and graphing software OriginPro, Version 2022, OriginLab Corporation, Northampton, MA, USA. Pearson’s correlation coefficient, r,
$$r=\frac{\left\{n\left(\sum xy\right)-\left(\sum x\right)\left(\sum y\right)\right\}}{\left\{\sqrt{\left\{\left[n\sum x^{\left\{2\right\}}–{\left(\sum x\right)}^{\left\{2\right\}}\right]\left[n\sum y^{\left\{2\right\}}–{\left(\sum y\right)}^{\left\{2\right\}}\right]\right\}}\right\}}$$
was used to calculate food waste composition variables and their effects on 2,3-BDO yields. Statistical significance was set at *p* < 0.05 and all statistical tests were performed open-source JavaScript components under different open-source licenses.

## 3. Results and Discussion

### 3.1. Food Waste Collection and Composition

Food waste samples were primarily composed of the cafeteria’s daily dining options (Table 1), with pizza, chicken, rice, mac and cheese, bread, and cake being found in multiple samples. Only 1 of the 13 samples was collected at breakfast time (Jul22) due to access to the cafeteria. Compositional analysis of the 13 samples was also carried out (Table 2), with the average moisture content at 34.7% ± 5.2%. This moisture content does not include the add water used create the food waste slurry and only takes into account the original moisture content from the collected waste. There is no standard moisture content for food waste and comparable studies show moisture content can range anywhere from 28% [28], 49% [29], and 95% [4] depending on the type and source of the food waste. The food waste solids content (on average) consisted of fat (19.7%), protein (18.7%), ash (4.8%), fiber (3.4%), starch (27.1%) and soluble sugars (20.9%) on a dry basis (Table 2). The average total mass balance equates to 94.65%, short of the expected 100%. This variance comes from a few samples (e.g., M31 and M24) which had very low mass balances of 75% and 86%, respectively, and is most likely due to other unmeasured food components such as soluble dietary fibers (e.g., pectin, inulin).

Fat composition was consistent with only two samples under 16% (Ju24 and M1) and 1 sample over 27% (Jul22) (Table 2). Protein content from the 13 samples showed only 1 sample (Ju24) under 10% of the total solids weight, which was expected as sample Ju24 was noted to be primarily fruit and bakery waste (Table 1). While ash and fiber content was low throughout all the samples (under 7%), starch was the largest single component. In sample M1, for example, starch (41.64%) was nearly half of all solids waste. This large starch composition is most likely attributed to the items listed in Table 1 (i.e., bread, mac and cheese, cake, pizza, and rice). These results are similar to other studies, where in one case, nutritional content of public primary school lunches resulted in carbohydrate (starch) values of 48.1% [30]. The remaining solids found in food waste were soluble sugars. The sugars measured were fructose, glucose, sucrose, and raffinose. Stachyose concentrations were also measured but were insignificant to mention. Of the four soluble sugars that were tested, glucose and sucrose were the two predominant sugars. The soluble sugars (especially glucose) found in food waste were highly variable, as seven samples had glucose values under 4% (F23, M1, M2, M23, M24, M31, A1), and five samples had glucose values above 15% (A27, Jul15, Jul24, Jul28, Jul20) of the dry weight. However, the five samples with higher glucose concentrations had lower starch concentrations (on average) than the seven samples with low glucose (under 4%). From this, we suspect that the natural starch degradation in food waste had progressed more in the samples with higher glucose (Figure 2) due to the fact that glucose is a main product of starch metabolism in microorganisms [31].

### 3.2. Fermentation and Production of 2,3-BDO 

From the 13 food waste samples, 2,3-BDO concentrations ranged from 4.05 g/L (sample A1) to 12.12 g/L (sample Ju24) (Figure 3). Fermentation samples were monitored and analyzed every 24 h for a duration of 72 h to see when the max 2,3-BDO was produced. It was found that 48 h was the optimum time to observe the highest 2,3-BDO concentrations. This was most likely due to the lower-than-expected free soluble sugars (glucose, sucrose and fructose) (Figure 2) causing the fermentation to stall in-between 48 and 72 h. Some of the highest amounts of 2,3-BDO production were, 12.12 g/L, 11.41 g/L, 10.09 g/L (Ju24, Ju28, Ju20, respectively) and the lowest 2,3-BDO concentrations were from food waste samples A1, F23, and Ju15 (4.05 g/L, 4.27 g/L, and 5.33 g/L, respectively) (Figure 3). The 2,3-BDO yields from carbohydrates (starch, glucose, sucrose, fructose, raffinose) were calculated and ranged from 13.2% (sample A1) to 55.2% (sample Ju28). The two highest recorded yields (55.2% and 51.6%) were slightly above the theoretical yield of 50% [14]. We attribute these over-theoretical values to the possibility that some lipids in food waste may have been converted to 2,3-BDO, thus increasing the yield based off carbohydrates alone. *Bacillus* sp. from other studies have shown to produce lipase during fermentation of waste cooking oil and future yield calculations may need to be adjusted to compensate for non-carbohyrdates [32]. Overall, the average yield from all 13 samples was 33% g/g of 2,3-BDO/carbohydrates. Under ideal conditions supplemented with glucose, yeast extract, and growth factors, *B. licheniformis* YNP5-TSU has been shown to have a maximum average yield of 0.46 g/g (92% of theoretical value) [23]. The average yield of 33% obtained from this study was a 13% reduction from the expect 46% yield under standard conditions [23]. In a similar study of open non-sterilized fermentation, *Enterobacter* sp. strain (FMCC-208) was able to yield 39% 2,3-BDO [33]. However, in this experiment 40 g of sucrose media was prepared and pasteurized (sterilized), before inoculation of strain FMCC-208. Many attempts have been made to modify strains in the Bacillaceae family to increase the 2,3-BDO yields, with the most notable attempts attaining yields of 42% with *B. licheniformis* MW3 [34] and *B. amyloliquefaciens* B10-127 with yields of 44% [35]. Other naturally 2,3-BDO producers have shown maximum yields of 47% (*B. licheniformis* 10-1-A) [36], 49% (*B. licheniformis* 24) [37], and 51% (*Paenibacillus polymyxa* ZJ-9) [38]. Since the average yields of 2,3-BDO by wild-type and genetically engineered strains are in 45–47% range, the 33% yield achieved in this study is comparatively low. However, the yields produced by *B. licheniformis* YNP5-TSU were from complex, untreated, food waste feedstocks without any outside added nutrients or sterilization of media. To our knowledge, this is the first study to claim yields this high from post-consumer food waste under these conditions. It is possible with the addition of external nutrients we could increase the yield of 2,3-BDO but whether this is economically feasible is something that needs to be further investigated.

### 3.3. Food Waste Composition and 2,3-BDO Correlation

To determine the most important factor influencing the 2,3-BDO yield, the Pearson correlation coefficient was calculated. As shown in Table 3, yields were compared with ash, fat, protein, starch, and glucose found in the 13 food waste samples. 

The percentage of glucose was the only component that had a positive correlation (r = 0.5654) with 2,3-BDO yields, and this correlation was significant (*p* < 0.05). This is an important factor for downstream fermentation into 2,3-BDO and is not surprising, as glucose is the key molecule to enter glycolysis in aerobic fermentation [39]. Through glycolysis glucose is reduced to pyruvate, and subsequently acetoin, which is produced by enzymes α-acetolactate synthase and α-acetolactate decarboxylase before final conversion to 2,3-BDO by 2,3-butanediol dehydrogenase [40]. All other components had a weak correlation and were not significant. Interestingly, though starch is a polymer of glucose, the Pearson coefficient for starch was r = 0.1581, indicating weak to no correlation. We suspect this is due to the fact that the YNP5-TSU *Bacillus* strain cannot effectively hydrolyze starch [22]. Our results were similar to a previous study by Poe et al. 2020, where high glucose concentrations in food waste led to a high butanol yield. Other studies [41,42,43] also showed converted lactic acid, fumaric acid, and ABE (acetone-butanol-ethanol) from food waste were all directly correlated to carbohydrate (glucose) concentrations. While growth factors, proteins, and phosphorous based molecules are important for bacterial growth, our first step is to increase future 2,3-BDO yields by pre-treating our food waste (e.g., starch and fiber) to increase initial glucose concentrations. Several studies have shown that pretreatment of food waste can indeed increase soluble sugar concentrations. Donzella et al. used pumpkin peel hydrolysate for lipid production and indicated concentrations of 52 g/L soluble sugars (glucose, sucrose, fructose) after pretreatment using Cellic CTec2 enzyme cocktail (hydrolytic activity > 1150 U/mL) [44]. In another valorization study, mixed food and beverage waste showed glucose (228.1 g/L) and fructose (55.7 g/L) were capable after saccharification with glucoamylase and sucrase for 12 h [45]. Even in the study where food waste was collected from food courts, a novel sequential acid-enzymatic hydrolysis process was able to increase the conversion efficiency of fermentable sugars by 85.38% based on the theoretical yields [46].

## 4. Conclusions

This study investigated the production of 2,3-BDO from non-sterilized post-consumer food waste. From this study we conclude that it is possible to produce 2,3-BDO, on average, at 33% yields (g/g of 2,3-BDO/carbohydrates), while simultaneously omitting traditional sterilization methods. This is significant since this reduces cost and makes the conversion of food waste more appealing by lowering energy consumption and processing time. Food waste collected in this study had sufficient nutrients to sustain growth for *B. licheniformis* YN5-TSU and on dry basis average, consisted of fat (19.7%), protein (18.7%), ash (4.8%), fiber (3.4%), starch (27.1%), and soluble sugars (20.9%). From our results we conclude that food waste is a potential feedstock that should be considered for bio-based 2,3-BDO. However, future research is needed to investigate food waste storage and expiration, pre-treatment options to increase initial soluble sugar concentrations, and addition of external growth factors for maximizing 2,3-BDO yields. 

## Figures and Tables

**Figure 1 foods-13-00452-f001:**
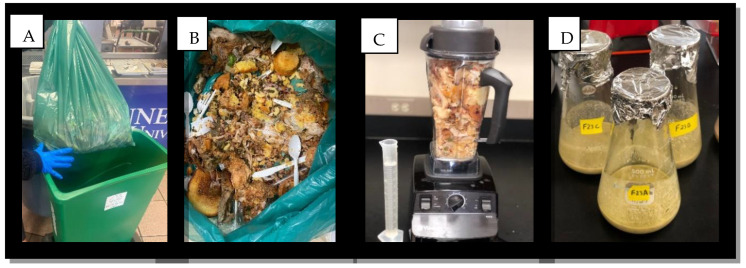
Raw food waste processing from the Dining Hall (Floyd Payne Student Center) at Tennessee State University, Nashville, TN. Food waste was collected by designated food waste receptacles within the cafeteria (**A**) and any non-fermentable items (i.e., plastics, paper) were sorted out (**B**). Food waste was combined with water and blended to a slurry (**C**) for sample preparation and downstream flask fermentation (**D**).

**Figure 2 foods-13-00452-f002:**
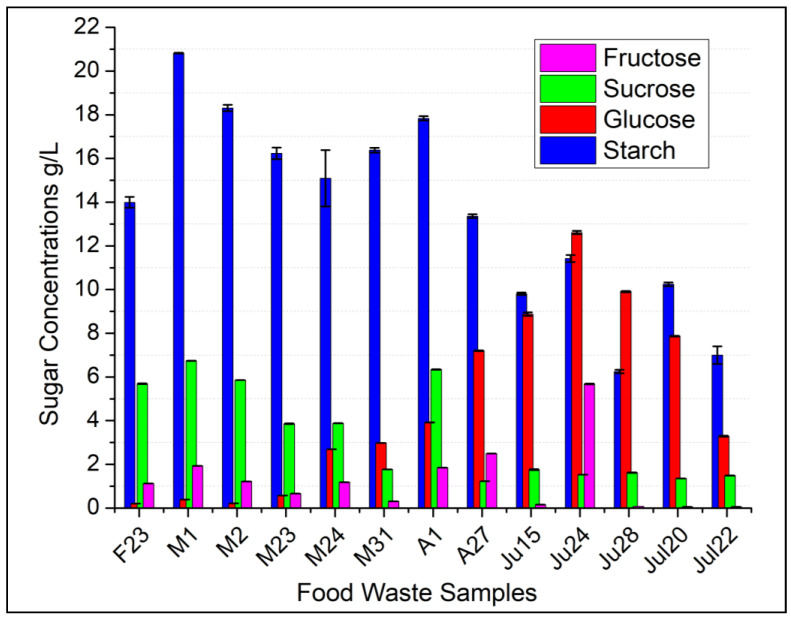
Most abundant carbohydrates found in food waste from 13 samples collected at the Tennessee State University dining hall in Nashville, TN.

**Figure 3 foods-13-00452-f003:**
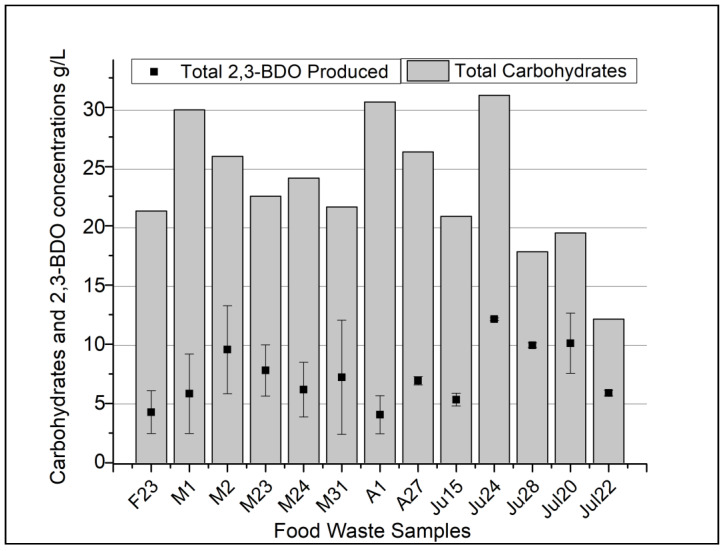
Total carbohydrates (g/L) from food waste and maximum 2,3-BDO produced during thermophilic non-sterile fermentation at 50 °C for 72 h.

**Table 1 foods-13-00452-t001:** Food Waste Sample Descriptions.

Sample Name	Sample Date	Known Food Waste Content
F23	23 February 2022	Fried and rotisserie-style chicken, rice, mac and cheese, collard greens, black bean burger, grilled cheese, French fries
M1	1 March 2022	Shrimp, crawfish, potatoes, corn, beans, rice, bread, cake
M2	2 March 2022	Fried chicken, mac and cheese, fresh salad, bread
M23	23 March 2022	Pizza, nachos, chicken, cake
M24	24 March 2022	Fried chicken, mac and cheese, rice, pizza
M31	31 March 2022	Fried Chicken, mac and cheese, toast, peas, rice, corn muffin, mixed vegetables
A1	14 April 2022	Rice, mixed vegetables, fried chicken, pizza, burgers, bread, corn, cookies
A27	27 April 2022	Pizza, burgers, corn muffin, grilled cheese, sub sandwiches, fresh salad
Ju15	15 June 2022	Chocolate cake, pizza, cantaloupe, peppers, bananas, hot dogs, rolls
Ju24	24 June 2022	Fruit, cabbage, pineapple, rice, cake
Ju28	28 June 2022	Rice, green beans, chips, chicken
Jul20	20 July 2022	Spaghetti, Brussel sprouts, rice, roast beef
Jul22	22 July 2022	Biscuit, eggs, sausage patties

**Table 2 foods-13-00452-t002:** Chemical composition of 13 food waste samples collected from campus dining halls.

		Dry Weight Composition					Soluble Sugars				
	Moisture Content Raw (%)	Fat (%)	Protein (%)	Ash (%)	Fiber (%)	Starch (%)	Fructose (%)	Glucose (%)	Sucrose (%)	Raffinose (%)	Mass Balance Total (%)
F23	29.12	26.60 ± 0.34	29.10 ± 0.10	4.75 ± 0.04	2.66 ± 0.08	27.98 ± 1.76	0.83 ± 0.05	0.45 ± 0.05	4.99 ± 0.27	0.85 ± 0.04	100.71 ± 2. 85
M1	29.33	11.24 ± 0.007	27. 14 ± 0.23	4.85 ± 1.01	7.03 ± 0.07	41.64 ± 0.07	1.31 ± 0.03	0.57 ± 0.18	2.18 ± 0.10	0.31 ± 0.06	100.35 ± 1.82
M2	33.62	18.57 ± 0.01	22.59 ± 0.19	4.82 ± 0.11	2.46 ± 0.43	36.61 ± 0.79	0.81 ± 0.10	0.58 ± 0.04	0.80 ± 0.01	0.95 ± 0.09	102.45 ± 1.12
M23	34.37	21.00 ± 0.09	21.47 ± 0.73	4.85 ± 1.01	3.94 ± 0.26	32.51 ± 1.62	1.32 ± 0.06	1.13 ± 0.06	7.71 ± 0.45	2.70 ± 0.15	94.13 ± 3.75
M24	33.34	22.19 ± 0.18	15.72 ± 0.39	4.82 ± 0.11	3.23 ± 0.67	30.18 ± 8.54	1.35 ± 0.03	1.14 ± 0.01	7.76 ± 0.20	2.78 ± 0.014	86.97 ± 10.14
M31	25.34	17.74 ± 0.02	12.44 ± 1.18	4.75 ± 0.04	4.09 ± 0.09	32.75 ± 0.651	0.61 ± 0.04	0.60 ± 0.05	3.55 ± 0.26	0.65± 0.11	75.00 ± 2.67
A1	35.00	20.67 ± 0.13	14.33 ± 0.86	2.43 ± 0.16	3.64 ± 0.18	35.67 ± 0.55	3.79 ± 0.06	3.38 ± 0.13	11.94 ± 0.21	1.50 ± 0.23	98.54 ± 2.56
A27	33.59	18.52 ± 0.19	14.19 ± 0.09	4.63 ± 0.02	1.75 ± 0.19	26.71 ± 0.68	5.28 ± 0.01	15.53 ± 0.30	2.45 ± 0.19	4.34 ± 0.20	99.6 ± 2.02
Ju15	38.53	16.63 ± 0.94	20.03 ± 0.40	4.74 ± 0.03	2.50 ± 0.06	19.61 ± 0.61	0.28 ± 0.08	17.76 ± 0.91	2.57 ± 1.512	0.75 ± 0.30	88.61 ± 4.96
Ju24	45.67	4.11 ± 0.36	9.13 ± 0.09	4.93 ± 0.09	6.53 ± 0.23	22.84 ± 1.40	12.56 ± 0.21	27.12 ± 0.60	3.19 ± 0.22	0.07 ± 0.02	99.08 ± 3.63
Ju28	38.76	25.58 ± 0.75	14.77 ± 0.23	6.83 ± 0.04	2.65 ± 0.11	12.50 ± 1.63	0.17 ± 0	19.99 ± 0.29	3.70 ± 0.58	0.19 ± 0.22	88.59 ± 3.99
Jul20	37.62	21.62 ± 0.87	26.11 ± 0.63	5.57 ± 0	2.73 ± 1.44	20.47 ± 0.81	0.12 ± 0.01	16.18 ± 0.18	2.75 ± 0.02	0.06 ± 0.01	98.07 ± 4.06
Jul22	37.11	31.18 ± 0.09	24.65 ± 0.35	4.93 ± 0.17	1.28 ± 0.05	13.99 ± 5.73	0.09 ± 0.02	7.12 ± 0.61	2.84 ± 0.26	0.79 ± 0.01	88.94 ± 7.34

**Table 3 foods-13-00452-t003:** Pearson correlation coefficient for food waste composition and 2,3-BDO yield.

2,3-BDO	Ash %	Fat %	Protein %	Starch %	Glucose %
*p*-Value at Significant Level *p* < 0.05	0.61	0.50	0.75	0.24	0.04 *
Pearson Coefficient (r)	−0.157	0.2024	−0.0981	0.1581	0.5654 *

* Indicates results are significant.

## Data Availability

Data is contained within the article.

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
