# Peer review of "Food Waste from Campus Dining Hall as a Potential Feedstock for 2,3-Butanediol Production via Non-Sterilized Fermentation"

_foods, 2024, doi:10.3390/foods13030452_

Round 1
Reviewer 1 Report
Comments and Suggestions for Authors
The essay presents a novel approach to utilizing food waste for 2,3-Butanediol production, which is significant in addressing environmental and economic concerns. The content is presented in a clear and scientifically sound manner, making it interesting to readers. Overall, the essay demonstrates good quality and provides valuable insights into the potential use of food waste as a feedstock for bioproducts.
1. The introduction provides sufficient background information on the issue of food waste and its impact on the environment. It also includes relevant references to support the statements made.
2. The cited references appear to be relevant to the research, providing support for the claims and findings presented in the essay.
3. The research design seems appropriate for the study, as it focuses on the production of 2,3-Butanediol from non-sterilized post-consumer food waste using microbial fermentation.
4. The methods are adequately described, including details on food waste collection, compositional analysis, culture propagation, fermentation, and quantification of 2,3-BDO and free sugars.
5. The results are clearly presented, including the composition of food waste samples, 2,3-BDO production, and correlations between food waste composition and 2,3-BDO yield.
6. The conclusions are supported by the results, with a clear link between food waste composition and 2,3-BDO production.
However, there are a few areas that could be revised to further improve the essay:
1. Clarity in the Introduction: The introduction provides a comprehensive overview of the issue of food waste, but it could be more focused. For instance, the essay could start with a concise statement of the problem, such as the environmental and economic impact of food waste, before transitioning into the specific research objectives related to 2,3-Butanediol production.
2. Detailed Method Descriptions: While the methods are adequately described, more detailed descriptions of specific steps, such as food waste collection, culture propagation, and fermentation, could enhance the reproducibility and clarity of the study.
3. Correlation Analysis: The essay briefly mentions the correlation analysis between food waste composition and 2,3-BDO yield, but it could benefit from a more detailed discussion of the implications and limitations of the findings.
4. Language and Style: While the overall quality of English language is good, minor improvements in language and style, such as sentence structure and word choice, could enhance the readability and flow of the essay.
Here are some examples:
· "This is different from previous studies where, agro-industrial waste, cheese, whey, molasses, dried fruit and vegetables, and bread waste were used to produce bioproducts before reaching the consumer." Suggested revision: "This differs from previous studies in which agro-industrial waste, cheese, whey, molasses, dried fruit and vegetables, and bread waste were used to produce bioproducts before reaching the consumer."
· "The 5 samples with higher glucose concentrations, however, had lower starch concentrations (on average) than the 7 samples with low glucose (under 4%)."Suggested revision: "However, the 5 samples with higher glucose concentrations had lower starch concentrations (on average) than the 7 samples with low glucose (under 4%)."
· "The percentage of glucose was the only component to have a positive correlation (r= 0.5654) with 2,3-BDO yields, which was significant (p < 0.05)." Suggested revision: "The percentage of glucose was the only component that had a positive correlation (r= 0.5654) with 2,3-BDO yields, and this correlation was significant (p < 0.05)."
· "The essay could benefit from a more detailed discussion of the correlation analysis between food waste composition and 2,3-BDO yield, including potential implications and limitations of the findings."Suggested revision: "The essay could benefit from a more detailed discussion of the correlation analysis between food waste composition and 2,3-BDO yield, including the potential implications and limitations of the findings."
· "The essay could include visual aids such as tables, graphs, or figures to illustrate the compositional analysis and 2,3-BDO production results, providing a clearer and more visual representation of the findings." Suggested revision: "The essay could include visual aids such as tables, graphs, or figures to illustrate the compositional analysis and 2,3-BDO production results, providing a clearer and more visual representation of the findings."
By addressing these examples and making the suggested improvements, the essay can be further enhanced in terms of clarity, detail, and overall impact.
Comments on the Quality of English LanguageLanguage and Style: While the overall quality of English language is good, minor improvements in language and style, such as sentence structure and word choice, could enhance the readability and flow of the essay.
Here are some examples:
· "This is different from previous studies where, agro-industrial waste, cheese, whey, molasses, dried fruit and vegetables, and bread waste were used to produce bioproducts before reaching the consumer." Suggested revision: "This differs from previous studies in which agro-industrial waste, cheese, whey, molasses, dried fruit and vegetables, and bread waste were used to produce bioproducts before reaching the consumer."
· "The 5 samples with higher glucose concentrations, however, had lower starch concentrations (on average) than the 7 samples with low glucose (under 4%)."Suggested revision: "However, the 5 samples with higher glucose concentrations had lower starch concentrations (on average) than the 7 samples with low glucose (under 4%)."
· "The percentage of glucose was the only component to have a positive correlation (r= 0.5654) with 2,3-BDO yields, which was significant (p < 0.05)." Suggested revision: "The percentage of glucose was the only component that had a positive correlation (r= 0.5654) with 2,3-BDO yields, and this correlation was significant (p < 0.05)."
· "The essay could benefit from a more detailed discussion of the correlation analysis between food waste composition and 2,3-BDO yield, including potential implications and limitations of the findings."Suggested revision: "The essay could benefit from a more detailed discussion of the correlation analysis between food waste composition and 2,3-BDO yield, including the potential implications and limitations of the findings."
· "The essay could include visual aids such as tables, graphs, or figures to illustrate the compositional analysis and 2,3-BDO production results, providing a clearer and more visual representation of the findings." Suggested revision: "The essay could include visual aids such as tables, graphs, or figures to illustrate the compositional analysis and 2,3-BDO production results, providing a clearer and more visual representation of the findings."
Author Response
Response for Reviewer 1
Thank you for your review and time spent helping to improve this manuscript. We have attached the edited manuscript for your reference with all changes made highlighted in yellow. Other colors are for other reviewers. Please view our following statements in connections to highlighted portions in manuscript with regards to your comments:
- Clarity in the Introduction: The introduction provides a comprehensive overview of the issue of food waste, but it could be more focused. For instance, the essay could start with a concise statement of the problem, such as the environmental and economic impact of food waste, before transitioning into the specific research objectives related to 2,3-Butanediol production.
Response: We have added another study discussing environmental impacts (line 19) and separated paragraphs to better the transition to microbial conversion.
- Detailed Method Descriptions: While the methods are adequately described, more detailed descriptions of specific steps, such as food waste collection, culture propagation, and fermentation, could enhance the reproducibility and clarity of the study.
Response: We appreciate this comment and have added several more details in the methodology. (lines 53-58,
- Correlation Analysis: The essay briefly mentions the correlation analysis between food waste composition and 2,3-BDO yield, but it could benefit from a more detailed discussion of the implications and limitations of the findings.
Response: We have extended our discussion on other related studies and yields obtained and how this compares to the yields in this study (lines 185-199)
- Language and Style: While the overall quality of English language is good, minor improvements in language and style, such as sentence structure and word choice, could enhance the readability and flow of the essay.
Response: The manuscript has been reviewed for grammar mistakes by colleagues proficient in English as their first language. It has also been edited for any spelling errors by all authors a second time. All examples of grammar issues provided by reviewer have been added into our new manuscript. We thank you for your time in these suggestions. Edits from reviewer examples: (line 37), (line 148), (line 210) other grammar edits: (line 17), (line 44), (line 30), (line 50) and other small errors have been fixed

Reviewer 2 Report
Comments and Suggestions for Authors
Please see the attachment.

Author Response
Response to Comments Reviewer 2
Thank you for your review and time spent helping to improve this manuscript. We have attached the edited manuscript for your reference with all changes made highlighted in light blue. Please view our following statements with regards to your comments:
- “This input of steam/heat increases the energy needed and severely reduces the profitability and sustainability of natural 2,3- BDO production. To circumvent sterilization needs a more recent approach has been shown effective and involves higher fermentation temperatures (above 45°C) using wild- type thermophilic bacteria, Bacillus licheniformis YNP5-TSU11.”
How can you prove your statement that a non-sterile fermentation at 50°C is more economical than a sterile fermentation at T=30°C e.g., for 2-3 days in a batch system, for 5-10 days in a fed-batch system or for 1-2 months in a continuous system?
Response: This is a very important question and we thank the reviewer for bringing this up. In a previous publication (https://doi.org/10.1016/j.fbp.2021.07.003) we briefly discuss the energy requirements of sterilization “This complex and highly sustainable nutrient waste must first be sterilized, typically to 121 °C by high pressure steam before being utilized in industrial fermentation. This step is time and cost consuming taking anywhere from 1 to 2 h and 853 J/mL per cycle (Li et al., 2018)” although not in this article. We did not conduct an economic analysis for length of time on different fermentation systems. We also do not have the food waste feedstock from this study any longer and therefore we have decided to re-word the sentence in the manuscript to “potentially” (see attached line 45-46).
- The first paragraph of 3.1 should be move to Materials and Methods.
Response: The first paragraph of 3.1 has been moved to the first part of 2.1 Methodology. (line 57-60 and 64-66)
- “This is different from previous studies where, agro-industrial waste, cheese, whey, molasses, dried fruit and vegetables, and bread waste were used to produce bioproducts before reaching the consumer”
Response: We agree with the addition of other ago-industrial feedstocks. We have added and referenced 6 other agro-industrial feedstocks (lines 38-43).
- Please cite and discuss more publication on non-sterile production of 2,3-BDO. There are indeed very limited, and the authors ought to have been aware of these very papers.
“A newly isolated Enterobacter sp. strain produces 2,3-butanediol during its cultivation on low-cost carbohydrate-based substrates. FEMS Microbiol. Lett. (2019), 366, fny 280.”
Response: We agree with more discussion on other 2,3-BDO non-sterile production and have added this into our discussion section along (referencing the paper mentioned) with a more detailed discussion of other wild-type and genetically modified strains and their yields from 2,3-BDO fermentation (lines 190-204). We feel placing this in the discussion section allows for a better comparison of our yields to other studies.
5.“Food waste media was prepared by adding 50 g of food waste into a Vitamix® 510 series blender with 50 ml distilled H2O in a 1:1 ratio and mixing until there was a 100mL homogenous mixture.”
How 50 g of a such variety of food waste was always yield a volume of 50mL? It would be more accurate to say that the concentration of the solution was 500g/L (namely, 50 g in 100 mL).
Response: We agree this is a better way to describe the food waste media preparation and have updated this in the manuscript (line 65-67).
- “After several attempts, we found the best consistency of food waste slurry (Figure 1C) for fermentation was created by adding 100 ml H2O to every 100 g of grinded food waste.”
The total quantity is different from that described in Materials and Methods (50ml + 50 g).
Response: We were referring to the 1:1 ratio of water to waste (100:100 and 50:50), however for clarification we have removed this line and condensed it in the methodology section (line 64-66).
- “This moisture content was back calculated to exclude the 50 ml DI H2O which was added to create the food waste slurry, since the solid waste was unable to be homogenized”
Here it is referred 50 mL of water.
Response: We have clarified this sentence and were meaning to state that the added water to create the slurry was not included in the moisture content calculation. (line 133-135)
- “After each fermentation period (0, 24 and 48 hours), 1 ml of the fermented samples was collected and centrifuged at 10,000 rpm for 10 min (Eppendorf© 5453 Minispin Plus Centrifuge)”
but in 3.2 and in the caption of Figure 3 you wrote
“Fermentation samples were monitored and analyzed every 24 hours for a duration of 72 hrs”
“Figure 3. Total carbohydrates (g/L) from food waste and maximum 2,3-BDO produced during ther- mophilic non-sterile fermentation at 50°C for 72 hrs.”
Response: This is a typo in our methodology section and it should read (0, 24, 48 and 72 hours), this has been updated (line 91)
